# Can we use sea surface temperature and productivity proxy records to reconstruct Ekman Upwelling?

Anson Cheung[1], Baylor Fox-Kemper[1], and Timothy Herbert[1]

[1]Department of Earth, Environmental, and Planetary Sciences, Brown University, Providence, RI 02912, USA

**Correspondence:** Anson H. Cheung (anson_cheung@brown.edu)

**Abstract.** Marine sediments have greatly improved our understanding of the climate system, but their interpretation often assumes that certain climate mechanisms operate consistently over all timescales of interest and that variability at one or few sample sites is representative of an oceanographic province. In this study, we test these assumptions using modern observations in an idealized manner mimicking paleo-reconstruction to investigate whether sea surface temperature and productivity proxy records in the Southern California Current System can be used to reconstruct Ekman upwelling. The method uses Extended Empirical Orthogonal Function (EEOF) analysis of covariation of alongshore windstress, chlorophyll and sea surface temperature as measured by satellites from 2002 to 2009. We find that EEOF1 does not reflect an Ekman upwelling pattern, but instead much broader California Current processes. EEOF2 and 3 reflect upwelling patterns, but these patterns are timescale dependent and are regional. Thus, the skill of using one site to reconstruct the large scale dominant patterns is spatially dependent. Lastly, we show that using multiple sites and/or multiple variables generally improve field reconstruction. These results together suggest caution is needed when attempting to extrapolate mechanisms that may be important on seasonal time scales (e.g. Ekman upwelling) to deeper time, but also the advantage of having multiple proxy records.

## 1 Introduction

The climate system varies across multiple timescales and is driven by both stochastic processes and deterministic forcings (Huybers and Curry, 2006). Paleoclimate records help us understand mechanisms of climate variability and change over long timescales by extending instrumental records beyond the historical period. Numerous studies have used paleoclimate records to understand climate system responses to different external forcings (e.g. Shakun et al., 2012), have put recent climate change into a long term context (e.g. Abram et al., 2016; PAGES2k Consortium, 2013), and have helped benchmark climate models (e.g. Harrison et al., 2015).

Marine sediment is one of the most widely used archives for paleoclimate studies. Using marine sediments for paleoclimate inference usually involves multiple steps, where one first measures multiple sensors, frequently proxies for sea surface temperature (SST) and productivity, from a single site. Then, one compares them with other nearby local records, hemispheric reconstructions, forcing reconstructions. Lastly, one applies modern large scale climatology to explain changes observed in paleoclimate records (e.g. Abram et al., 2016; Goni et al., 2006; Leduc et al., 2010a; MARGO, 2009; McGregor et al., 2007; Vargas et al., 2007). While these comparisons have improved our understanding about paleoclimate significantly, uncertain-

ties and oversimplifications often may result in overly broad interpretations and assertions. Notably, this approach typically assumes that (1) certain climate mechanisms always operate over the past at all timescales of interest, and (2) large scale phenomena can be linked to variability at one or a few sample sites (i.e., a paleoclimate record location). In actuality, some have found substantial difference in SST reconstruction at nearby sites (e.g. Leduc et al., 2010b, and references therein).

This paper illustrates an approach to test commonly asserted interpretations of SST and productivity proxy records by using observational data to analyze a region where a known mechanism drives a large fraction of the variability, and with well preserved high resolution sedimentary records – the southern California region, an example of an Eastern Boundary Upwelling Systems (EBUS). There are strong scientific and societal interests to understand EBUS because physical and biogeochemical changes in these regions are known to have significant impacts on regional climate (Snyder et al., 2003; Ravelo et al., 2004;

Jacox et al., 2014) and global fishery industry (Ryther, 1969; Pauly and Christensen, 1995; Ware and Thomson, 2005). Unfortunately, it remains uncertain how EBUS will change on decadal to centennial timescales in the future (Bakun et al., 2015; Di Lorenzo, 2015; Garcia-Reyes et al., 2015, and references therin). Nevertheless, underlying sediments in these regions often accumulate rapidly and contain a wealth of paleoclimate information, in particular organic biomarkers and associated proxies. Thus, this has allowed high resolution (subdecadal timescale) and high quality paleoclimate reconstructions along many

EBUS, which provide additional constraints on past and future changes of EBUS (e.g. Leduc et al., 2010a; McGregor et al., 2007; Vargas et al., 2007).

Variability in SST and productivity reconstructions along EBUS are often regarded as a response to Ekman pumping (e.g. Leduc et al., 2010a; McGregor et al., 2007; Vargas et al., 2007; MARGO, 2009). However, many other processes are also at play in EBUS and can drive SST and productivity changes (e.g. eddies, zonal advection, surface heat flux variations, changes

in nutrient sources and concentration forced by subsurface processes, and large-scale climate variability that affects the stratification) (Di Lorenzo et al., 2005; Chhak and Di Lorenzo, 2007; Gruber et al., 2011; Jacox et al., 2016; Rykaczewski and Dunne, 2010; Xiu et al., 2018). Depending on spatial and temporal timescales, these processes can overwhelm the Ekman signal in SST and productivity changes recorded by proxy records.

Here we use high resolution modern observations available during the satellite era, to probe the spatial and temporal influ-

ence of Ekman pumping on environmental parameters of interest (e.g. SST and productivity). We apply the Extended Empirical Orthogonal Function (EEOF) approach (Chen and Harr, 1993) to analyze covariation between sea surface temperature, productivity, and alongshore wind stress in the Southern California Current System using high resolution satellite data. We test the hypotheses that (1) the dominant covarying EEOF pattern resembles region-wide Ekman upwelling, (2) Ekman upwelling patterns, and thus the wind-stress magnitude, can be recovered using time-averaged proxies, and (3) large-scale changes are

not the dominant drivers of variability at a single paleoclimate site. We also assess the benefits of using multiple proxy records from multiple sites to better understand climate variability of the past in EBUS regions.

## 2 California Current System

The availability of numerous high resolution spatiotemporal data (e.g. repeated hydrography, gliders, satellites) and advances in modeling have allowed us to better understand variability of the California Current System (CCS) on multiple timescales. The CCS is made up of the California Current, California Undercurrent, and upwelling zones, which interact with a variety of local

topographic features and estuaries. On first order, the CCS, as a whole, is driven by large-scale climate forcing. Changes in atmospheric pressure systems (subtropical high, Aleutian low) alter wind strength and direction, which in turn affects currents' direction, strength and upwelling variability. The stratification in the region is set by large-scale features and forcing of the North Pacific. Variations in topographic features, wind forcing, freshwater inputs, and submesoscale-mesoscale features across spatial scales also play important roles in determining spatiotemporal characteristics of the CCS. Lynn and Simpson (1987);

Checkley Jr and Barth (2009); Capet et al. (2008) provide overviews on dynamics of CCS and drivers of SST, chlorophyll, and wind forcing variability.

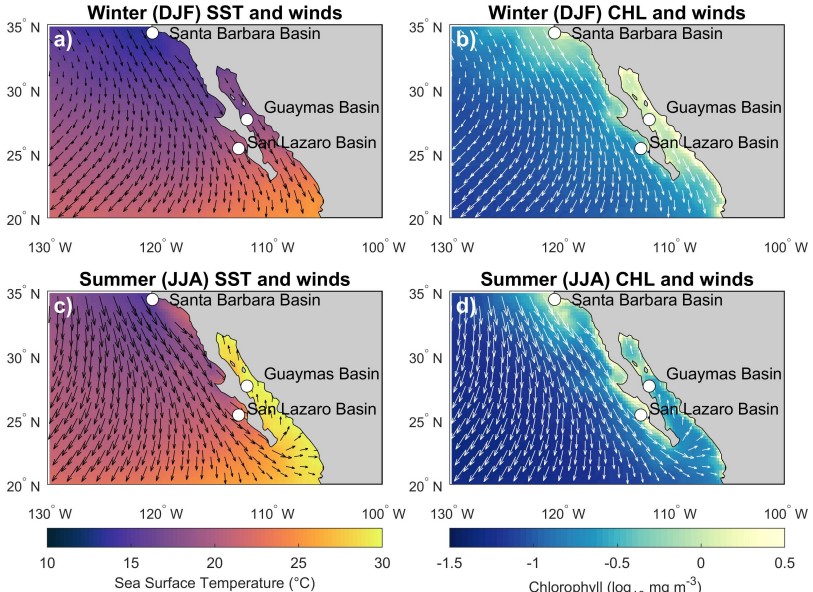

**Figure 1.** a) Winter (December, January, February) sea surface temperature average and wind pattern; b) Winter chlorophyll monthly average and wind pattern; c) Summer (June, July, August) sea surface temperature average and wind pattern; d) Summer chlorophyll monthly average and wind pattern. The basins highlighted are where high-resolution (subdecadal) sediment cores were previously retrieved and analyzed.

The optimal marine sediments to reconstruct subdecadal climate variability require high sedimentation rate with minimal bioturbation, and hence anoxic depositional environments. Along the CCS, these conditions mostly occur south of $24°N$ with exceptions of silled basins (e.g. Santa Barbara Basin) (van Geen et al., 2003). As a result, previous high resolution (subdecadal)

paleoclimate studies were mostly done in the Southern part of the CCS (SCCS; Fig. 1) (e.g. Goni et al., 2006; Abella–Gutiérrez and Herguera, 2016; Zhao et al., 2000).

## 3   Data and Method

This study made use of high spatiotemporal resolution estimates of sea surface temperature (SST), chlorophyll-a (CHL) and alongshore wind stress (TAU) from satellite measurements to assess the role of Ekman pumping in driving SST and productivity changes along SCCS. We used Extended Empirical Orthogonal Function (EEOF) to assess the covariation between these variables because they are expected to be correlated spatially and temporally if Ekman theory is indeed the primary mechanism driving changes in the region. EEOF analysis decomposes the dataset into different covarying patterns that are orthogonal to each other. Each covarying pattern is accompanied by a timeseries that represents the time evolution of the covarying pattern. These patterns do not necessarily correspond to dynamical modes, but they are suggestive of physical processes that are present in the system (Monahan et al., 2009). Thus, analysis on EEOF patterns allow us to make inference about the potential underlying dynamics. In addition, we assessed the effects of time averaging and spatial subsampling on the ability to recover dominant patterns within the spatial window analyzed. Such assessment allows us to determine the fidelity of using proxies, which are time averaged and undersampled spatially, to understand Ekman pumping in SCCS. Details of the data and method used can be found in sections 3.1 and 3.2.

### 3.1   Data

We used sea surface temperature (SST) from GOES, chlorophyll-a (CHL) from MODIS, and alongshore wind stress (TAU) observations from QuikSCAT that span from July 2002 to November 2009. Although CHL does not equate precisely to primary productivity, and also differs from productivity inferred from proxy records, CHL provides a first order estimate of productivity (Henson et al., 2010). All data were derived and available from National Aeronautics and Space Administration Jet Propulsion Laboratory PO.DAAC and ocean color data server. We did not use the California Cooperative Oceanic Fisheries Investigations dataset because sampling resolution is low and the spatial extent is small when compared to satellite images. Reanalysis products (e.g. SODA) were also not chosen because even though they may span a longer period of time, there are many uncertainties associated with these products, for instance initial conditions, boundary forcings, model physics and resolution (approx 25km horizontal) (Carton et al., 2018). Furthermore, Capet et al. (2008) show that submesoscale-permitting resolution (750m horizontal) is needed in order to accurately simulate this upwelling system.

For TAU calculation, we used the descending pass of level 3 gridded Jet Propulsion Laboratory v2 QuikSCAT surface wind observations (SeaPAC, 2006). The QuikSCAT satellite is equipped with the SeaWinds scatterometer, a microwave radar that measures ocean radar backscatter over a cross section, which varies with satellite parameters and surface geometry (Freilich et al., 1994; Chelton and Freilich, 2005). Surface wind vectors can be estimated using model functions to estimate the relationship between wind and radar backscatter over the cross section. Level 3 data were derived using the Direction Interval Retrieval with Threshold Nudging wind vector solutions based on Level 2B data, which used the QSCAT-1B geophysical model function (Perry, 2001). Level 3 QuikSCAT data provides $0.25° \times 0.25°$ spatial resolution on a daily timescale. The QuikSCAT accuracy is about $0.75\ \mathrm{ms}^{-1}$ in the along-wind component and about $1.5\ \mathrm{ms}^{-1}$ in the crosswind component (Chelton and Freilich, 2005).

We utilized SST observations from Geostationary Environmental Satellites (GOES). GOES satellites provide near-time SST measurements along the west coast of North America. We used level 3 gridded GOES 6km near time real SST daily data after May 12, 2003 (NOAA/NESDIS, 2003b) and averaged hourly SST data to daily mean resolution prior to May 12, 2003 (NOAA/NESDIS, 2003a). Level 3 GOES SST data provides $0.05° \times 0.05°$ spatial resolution with better than 1K SST accuracy

(Wick et al., 2002).

For CHL concentrations, we used ocean color from the Moderate Resolution Imaging Spectroradiometer on the Aqua satellite (MODIS-Aqua) (Hu et al., 2012). MODIS-Aqua is sun synchronous and measures 36 spectral bands. We used level 3 standard mapped image CHL measurements from MODIS-Aqua v2014.0 (O.B.G.P., 2015). Level 3 CHL data provides $0.041° \times 0.041°$ spatial resolution on near daily timescale with an accuracy of approximately $\pm35\%$ (Dall'Olmo et al., 2005).

## 3.2  Method

### 3.2.1  Observation Pre-processing

To allow comparison between SST, CHL, and TAU, CHL and SST were regridded to $0.25° \times 0.25°$ spatial resolution. This was done by bounding the datasets to $15° - 45°N, 130° - 100°W$, then calculating the area weighted CHL and SST value over each new grid cell. We further restricted our latitudinal extent to $15° - 35°N$ to make the analysis more computationally

efficient. Repeated analysis using different spatial domains (case 1: only east of $125°W$; case 2: only north of $20°N$) suggests our conclusions are insensitive to the spatial extent selected for analysis (not shown).

Since the primary interest is Ekman driven upwelling along the coast, we computed the TAU by using (1):

$$\tau = \rho C_D U |\boldsymbol{U}| \tag{1}$$

where $\tau$ = alongshore wind stress, $\rho$ = air density, $C_D$ = drag coefficient, $U$ = wind speed, $|\boldsymbol{U}|$ = alongshore wind vector. $|\boldsymbol{U}|$

was calculated by summing the alongshore component of zonal and meridional wind vectors such that $-|\boldsymbol{U}|$ and $|\boldsymbol{U}|$ represent equatorward and poleward wind stress respectively. We used constant values for the coefficients, where $\rho = 1.2kg/m^3$ and $C_D = 1.2 \times 10^{-3}$ (Large and Pond, 1981).

Linear interpolation of all of the near-daily datasets temporally ensured uniform daily sampling rate data at each grid cell. The logarithm of CHL data was taken after regridding but before EEOF analysis because CHL exhibits a nearly log-normal

distribution (Campbell, 1995). Before EEOF analysis, each variable was normalized by dividing the dataset by its domain-wide and all-time standard deviation, which makes the anomaly variations in each variable comparable to each other in terms of occurrence likelihood (assuming approximately Gaussian distributions).

To follow the logic of analyzing fields that would resemble proxy records, no removal of mean or climatological states or seasonality from the satellite records was performed. Thus, the EEOF analysis is performed on the total fields, rather than the

anomaly fields.

### 3.2.2 Extended Empirical Orthogonal Function

Extended Empirical Orthogonal Function (EEOF) decomposition analysis was used to extract dominant patterns with covariation in SST, CHL, and TAU. EEOF is a variant of Empirical Orthogonal Function (EOF) analysis, a method that extracts coherent, orthogonal patterns by optimizing variance into multiple orthogonal functions in time and space. Multiple variants of the EOF exist, which all involve taking into account temporal correlations of a variable or correlations between variables (e.g. Bretherton et al., 1992; Hannachi et al., 2007, and reference therin). Examining multiple time snapshots as a single field allows EOF-based analysis to extract propagating patterns (e.g. Chen and Harr, 1993) and covarying patterns (Kutzbach, 1967). Figures 2 and 3 show examples of three different EOF based methods that are fundamental to the analysis herein (EOF, EEOF (temporal correlation), EEOF (multiple variables)).

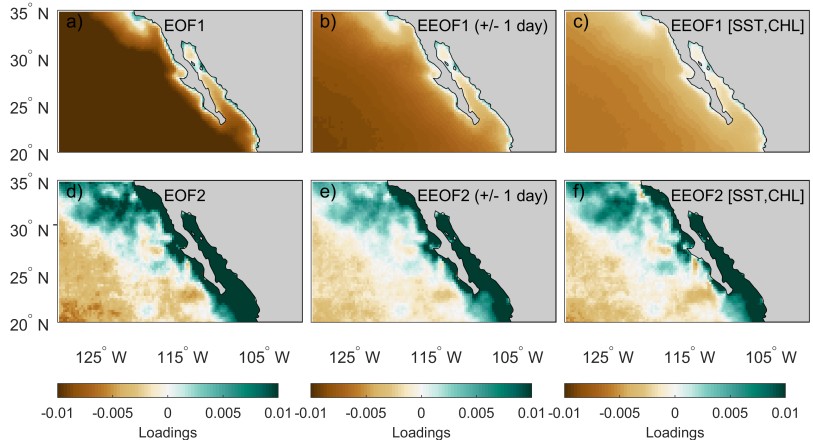

**Figure 2.** Example of decomposing sea surface temperature into different modes by using an (a, d) empirical orthogonal function, (b, e) an extended empirical orthogonal function with 1 day lead and lag, and (c, f) an extended empirical orthogonal function with chlorophyll included.

The EEOF method used in this study involved extracting dominant covarying patterns by taking into account both temporal correlation within the same variable (symmetric lead-lag relationships) and correlation between variables. We employed the singular value decomposition (SVD) method (Bretherton et al., 1992) to decompose the covarying pattern of SST, CHL, and TAU into the relevant EEOF objects.

To consider time correlation of a variable $X$ for EEOF analysis, we form the following data matrix:

$$X = \begin{pmatrix} x_{1,1} & \cdots & x_{1,j} & x_{1+k,1} & \cdots & x_{1+k,j} & x_{1+2k,1} & \cdots & x_{1+2k,j} \\ \vdots & \ddots & \vdots & \vdots & \ddots & \vdots & \vdots & \ddots & \vdots \\ x_{m-2k,1} & \cdots & x_{m-2k,j} & x_{m-2k+1,j} & \cdots & x_{m-2k+1,j} & x_{m,1} & \cdots & x_{m,j} \end{pmatrix} \tag{2}$$

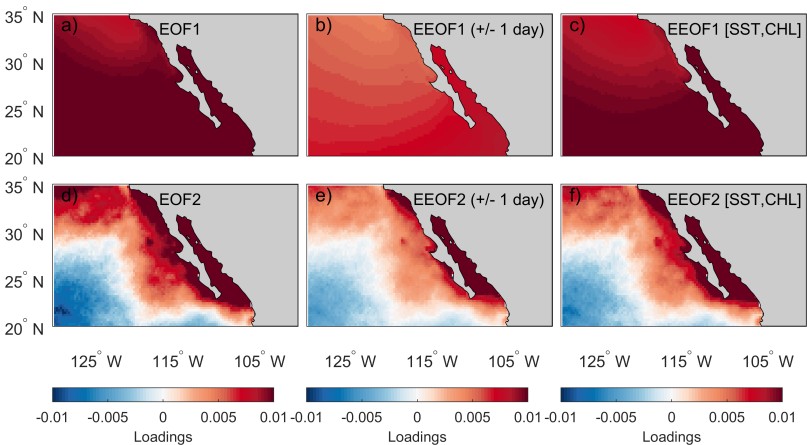

**Figure 3.** Example of decomposing chlorophyll into different modes by using an (a, d) empirical orthogonal function, (b, e) an extended empirical orthogonal function with 1 day lead and lag, and (c, f) an extended empirical orthogonal function with sea surface temperature included.

Where $x_{t,i}$ is a data point at a certain time snapshot t and space gridpoint i, $t = 1, 2, ..., m$, $i = 1, 2, ..., j$, $k =$ time unit of lead and lag included, $m =$ temporal length of the dataset, and $j =$ total spatial grid points covered. Thus, $X$ is the concatenation of multiple reproductions of $x_{t,i}$, with each column featuring $x$ evaluated at sequential times, and each row representing every spatial value of $x$, concatenated with spatial maps that are displaced in time to provide lead and lag information. Similarly, the
data matrix, $M$, with three variables can be written as follow:

$$M = \begin{pmatrix} SST & | & CHL & | & TAU \end{pmatrix} \tag{3}$$

Where SST, CHL, and TAU are submatrices with structure similar to matrix $X$. Note that each row of $M$ is a complete spatiotemporal set of each variable, including every spatial location and lead and lag times for each variable, so that $M$ is effectively the concatenation of three $X$ matrices, one for each variable. Then, using SVD, we can decompose (3) into:

$$M = USV^T \tag{4}$$

where $U = $ a matrix of left orthogonal, singular vectors as columns with temporal information of the M matrix (Principal Components (PCs)), $S = $ singular values, and $V = $ a matrix of right orthogonal, singular vectors as columns with spatial information (Extended Empirical Orthogonal Functions (EEOFs)) of the M matrix. Note that the SVD method arrives at a basis of eigenvectors of the covariance matrices $M^T M$, i.e., $M^T MV = S^2 V$, and $MM^T$, i.e., $MM^T U = S^2 U$, so this approach
is equivalent (although slightly different algorithmically) to generating EEOFs by eigenvalue decomposition.

Since proxy records reflect time averaged environmental information (usually monthly or longer), daily satellite information for analysis does not accurately depict the temporal smoothing characteristics in proxies. Hence, we performed EEOF analysis

independently on daily data after averaging it into 30 days ($\sim$ monthly), and 365 days ($\sim$ annual) with non-overlapping means. The relatively short span of satellite observations does not allow us to extend our analysis to longer time periods that might also be of interest.

### 3.2.3 Determining the significance of modes and lead-lag

5    Based on singular values, EEOF1 explains $\sim 85\%$ of the total variance, EEOF2 and 3 each explains $\sim 5\%$ of the total variance. Instead of using singular values to determine the significance of each mode, we selected the number of modes and lead-lags to retain by evaluating the skill to reconstruct TAU. This approach was motivated by the interest of this study to detect Ekman upwelling, which involves covariation of SST, CHL, and TAU, and our inability to reconstruct TAU directly using proxies. Reconstruction of TAU ($TAU_{rec}$) was carried out as follow:

$$M_0 = \Big( SST \quad | \quad CHL \quad | \quad TAU(0) \Big) \tag{5}$$

where $TAU(0)$ = the columns for TAU in the original data matrix were replaced with zeros. Then,

$$M_{rec} = rM_0V_nV_n^T \tag{6}$$

$$M_{rec} = \Big( SST_{rec} \quad | \quad CHL_{rec} \quad | \quad TAU_{rec} \Big) \tag{7}$$

where $r$ = rescaling factor calculated by $\frac{std(SST|CHL)}{std(SST_{rec}|CHL_{rec})}$, $V_n$ = spatial information obtained from decomposing M (Eq. 4)

15    with n numbers of mode retained, where n=1...5. Note that as $n$ is much smaller than the rank of $M_{rec}$, $V_nV_n^T$ is not the identity matrix, but is better thought of as the projection of $M_0$ onto the leading modes of $M$. As zero wind stress is inconsistent with any of the modes $V_n$, multiplying $M_0$ by this factor adds $TAU$ variability back into the zeroed values that is more consistent with the observed $SST$ and $CHL$, which is $M_{rec}$.

   We used Root Mean Square Error (RMSE) as a metric to measure agreement between reconstructed TAU and actual TAU:

$$RMSE = \sqrt{\overline{(TAU_{rec} - TAU)^2}} \tag{8}$$

where $\overline{[\cdot]}$ represents mean of the data.

   Our analysis shows that reconstruction using three modes with no lead-lag information included provides the most stable result in predicting TAU from SST and CHL regardless of averaging timescale (Fig. 4). This result, and similar results of convergence accuracy by adding more modes, suggest that the first three modes ($n = 3$) are reliable in this and other analyses,

25    which will be used for the remainder of this paper.

### 3.2.4 Reconstruction of Principal Components

We determined how well proxy records could represent large scale circulation patterns by means of signal reconstruction. We focused specifically on three sites with previously published high-resolution paleoclimate records – Santa Barbara Basin,

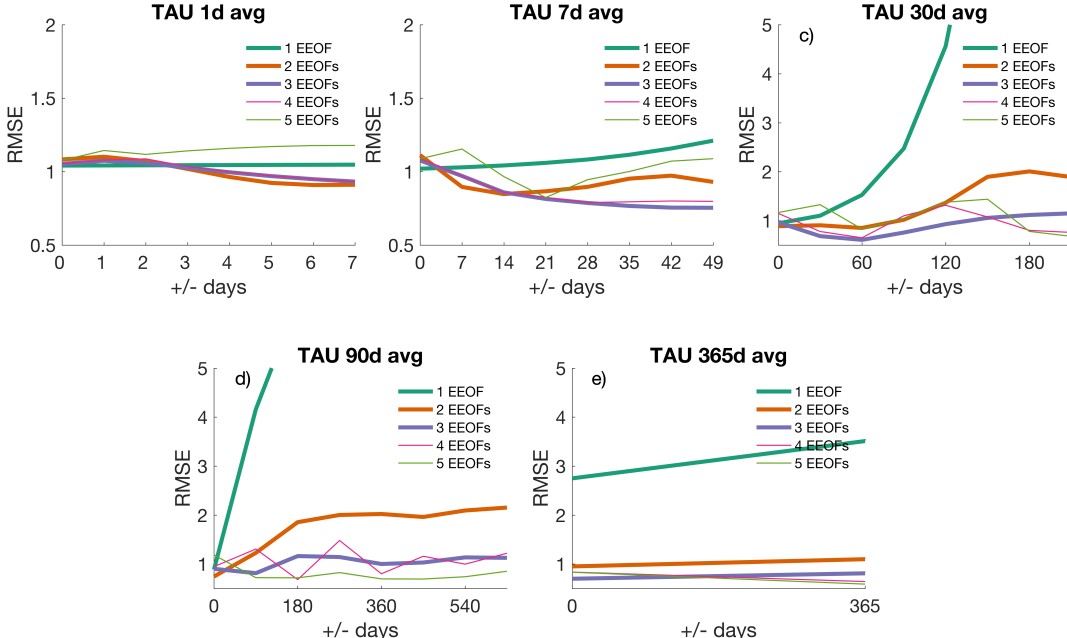

**Figure 4.** Root Mean Square Error of reconstructed wind stress with respect to actual wind stress using a) daily data; b) 7-day averaged data; c) 30-day averaged data; d) 90-day averaged data; e) 365-day averaged data.

San Lazaro Basin, and Guaymas Basin – and two environmental variables, SST and productivity (Goni et al., 2006; Abella–Gutiérrez and Herguera, 2016; Zhao et al., 2000). We carried out three different kinds of reconstructions to address (1) how well does a single site/proxy record represent large scale circulation? (2) Does increasing the number of proxy records and/or sites improve the skill to represent modes extracted from EEOF analysis? and (3) Does increasing proxy records and/or sites improve the skill to reconstruct the original dataset? This was achieved by first only retaining the target time series (i.e., those proxy records that are to be included) from the location in $M_{tar}$:

$$M_{tar} = \begin{pmatrix} 0 & \cdots & \cdots & tar_{1,j} & \cdots & 0 \\ \vdots & \ddots & \vdots & \vdots & \ddots & \vdots \\ 0 & \cdots & \cdots & tar_{m,j} & \cdots & 0 \end{pmatrix} \tag{9}$$

We reconstructed the temporal evolution of each mode by (10), using only the targeted proxy records and $n$ EEOF modes:

$$U_{rec} = r_s M_{tar}(S_n V_n^T)^{-1} \tag{10}$$

We reconstructed the dataset by (11):

$$M_{rec} = r_s M_{tar} V_n V_n^T = U_{rec} S_n V_n^T V_n V_n^T \tag{11}$$

where $U_{rec}$ = reconstructed PCs, $r_s$ = ratio between the standard deviation of timeseries from target site(s) and the standard deviation of the reconstructed timeseries from target site(s), $S_n$ and $V_n$ were derived from (4) and $n$ = modes retained for analyses. In this scenario, only the parts of $V_n$ associated with the target location were retained for reconstruction. The pseudo-inversion of the matrix $S_n V_n^T$ was done using Moore-Penrose pseudo-inverse, which amounts to inverting only the non-singular degrees of freedom, while zeroing out the remaining modes. Similarly, the multiplication of $M_{tar}$ by $V_n V_n^T$ considers only the projection of $M_{tar}$ onto the $n$ retained modes ($VV^T$ is the identity matrix, but if only some modes are retained, then only $V_n^T V_n$ is an identity, but over the smaller dimensional space spanned by the retained modes). By retaining 1 mode ($n = 1$) and limiting the proxy record used in $M_{tar}$ to 1, equations 10 and 11 can be used to addressed the ability of using a proxy record at a single location to represent large scale circulation, which is represented by EEOF1. Similarly, by retaining 3 modes ($n = 3$), equations 10 and 11 can be used to evaluate the effects of increasing proxy records to reconstruct modes extracted from EEOF analysis and the original dataset.

## 4   Results and Discussion

### 4.1   Does the dominant covarying pattern reflect Ekman upwelling?

EEOF analysis on daily resolution data displays spatial patterns that are distinct from what would be expected from Ekman upwelling. By computing cross-shore (the difference divided by its arc length at locations: $25.375°$N,$-112.875°$W and $22.875°$N,$-120.625°$W) and meridional gradients (the difference divided by its arc length at locations: $34.375°$N,$-120.625°$W and $22.875°$N,$-120.625°$W) and comparing them, we find TAU and CHL display a weak cross-shore gradient compared to their own respective meridional gradient. On the other hand, SST exhibits a meridional gradient that is stronger than its cross-shore gradient (Fig. 5). These patterns remain dominant when 30-day and 365-day averaged data were used.

The fact that EEOF1 does not resemble Ekman upwelling pattern has two major implications. First, this implies that wind stress is not the only forcing that drives CHL and SST changes along EBUS. Previous studies have reported different mechanisms that could control changes in CHL or SST along EBUS on various timescales. For instance, changes in subsurface nutrient concentration and sources have shown to alter primary productivity (Chhak and Di Lorenzo, 2007; Rykaczewski and Dunne, 2010) whereas surface heat flux has shown to exert dominant control on sea surface temperature in the California Current System (Di Lorenzo et al., 2005). Our study confirms these results and further iterates the importance of considering different factors that could affect CHL and SST along EBUS, which are often used as indicators for changes in Ekman-driven upwelling. Second, paleoclimate reconstructions in the SCCS will be unlikely to reflect Ekman upwelling, in contrast to the common paradigm in the field, and couplings observed between proxy reconstructions of e.g. SST and productivity likely capture other processes.

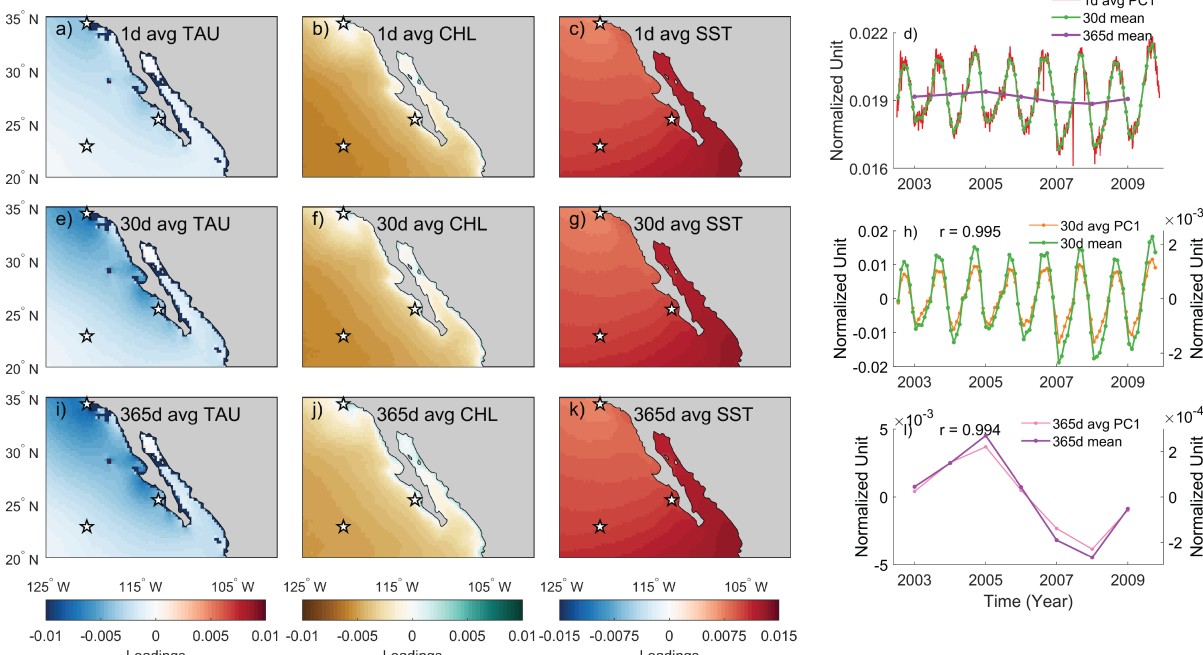

**Figure 5.** EEOF1 spatial and temporal patterns of TAU, CHL, and SST using a–d) daily; e–h) 30 day averaged; i–l) 365 day averaged data. Stars in spatial pattern plots indicate locations where the differences were taken to compute crossshore and meridional gradients. 30 day mean (green) and 365 day mean (purple) timeseries were derived from averaging 1d avg PC1. Correlation coefficient indicates how well does time mean of 1d avg PC track PC of time averaged data.

## 4.2 Can time-averaged proxies be used to reconstruct Ekman upwelling?

Even though the dominant covarying pattern does not reflect Ekman upwelling, the EEOF method allows us to decompose multiple covarying patterns for analysis. Our results suggest that EEOF2 and EEOF3 resemble Ekman upwelling pattern on daily timescales, but they reflect upwelling at different locations (Figs. 6 – 7). Specifically, EEOF2 depicts upwelling conditions

5 off Baja California whereas EEOF3 reflects upwelling or other rapid change in conditions at Sea of Cortez. This presents an opportunity to understand whether time averaged proxies can be used to reconstruct Ekman upwelling, given optimal site selection.

Visual comparison of EEOF2 and EEOF3 patterns across different averaging windows suggest these patterns change with respect to the averaging window. For both EEOFs, their patterns resemble Ekman upwelling when data with daily resolution is

10 used. These Ekman-upwelling-like patterns disappear when 365 day averaged data is used instead and only spatially incoherent structures are retained (bottom rows of Figs. 6 – 7). The disappearance of an Ekman upwelling pattern suggests that either Ekman upwelling is a subannual process and/or that this process is not a dominant feature on an annual timescale. We further analyze the changes in temporal scale by comparing 30 day and 365 day averages of the principal component derived using

daily data with principal components derived from 30 day and 365 day average data. The averages of the principal component derived using daily data represent the assumption that the same dynamical process happen at all timescales whereas the principal components derived from averaged data represent the actual covarying pattern on the timescale of interest. Our results show that 30 day and 365 day mean of PC2 and PC3 derived from daily data do not always track the principal components derived from time averaged data (Figs. 6h, l and 7h, l). While it is not possible to diagnose the underlying cause using our method, these results imply that marine sedimentary records, which generally integrate over the annual cycle, cannot capture Ekman upwelling variations in this region. Furthermore, these results highlight the importance of considering what timescales are reflected in the proxy record. On the assumption that some proxies are seasonally biased (e.g. "integrated production temperature" applied to the interpretation of alkenone paleotemperature estimates by (Conte et al., 1992)) we add a sine weighting function (maximum in March and minimum in Sept) to the 30 day averaged dataset and reanalyze the resulting EEOF pattern. We find that the pattern is similar to the one without weighting (not shown). This suggests that the seasonal cycle does not dominate the resulting EEOF patterns over this spatial and temporal domain.

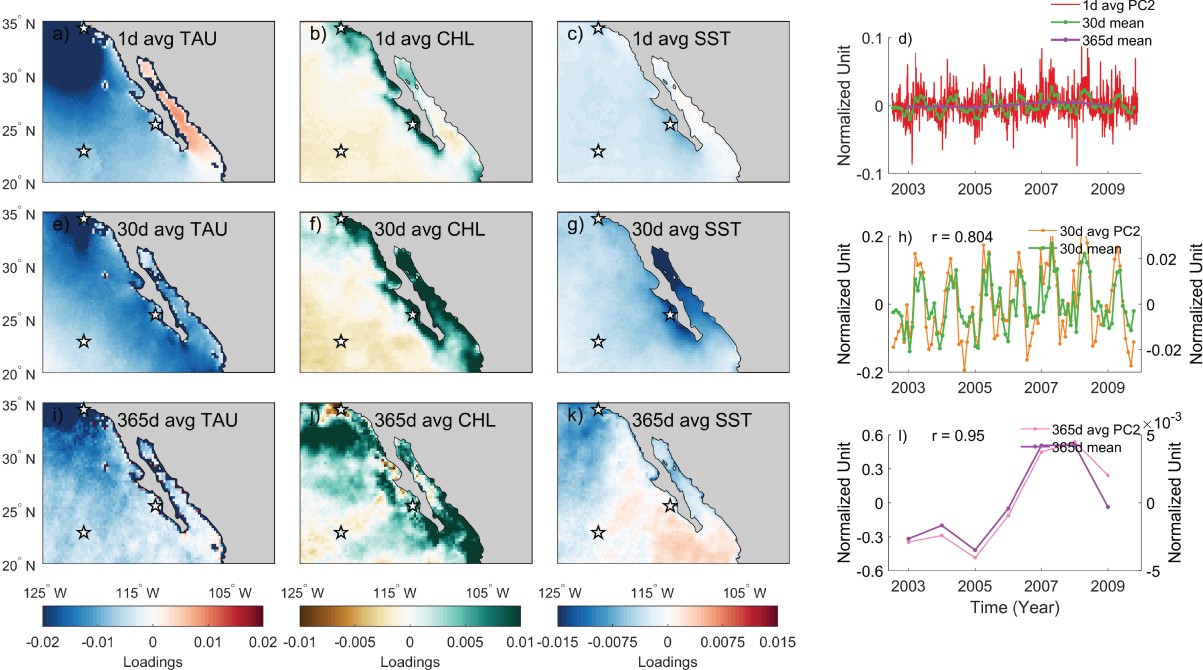

**Figure 6.** EEOF2 spatial and temporal patterns of TAU, CHL, and SST using a–d) daily; e–h) 30 day averaged; i–l) 365 day averaged data. Stars in spatial pattern plots indicate locations where the differences were taken to compute crossshore and meridional gradients. 30 day mean (green) and 365 day mean (purple) timeseries were derived from averaging 1d avg PC2. Correlation coefficient indicates how well does time mean of 1d avg PC track PC of time averaged data.

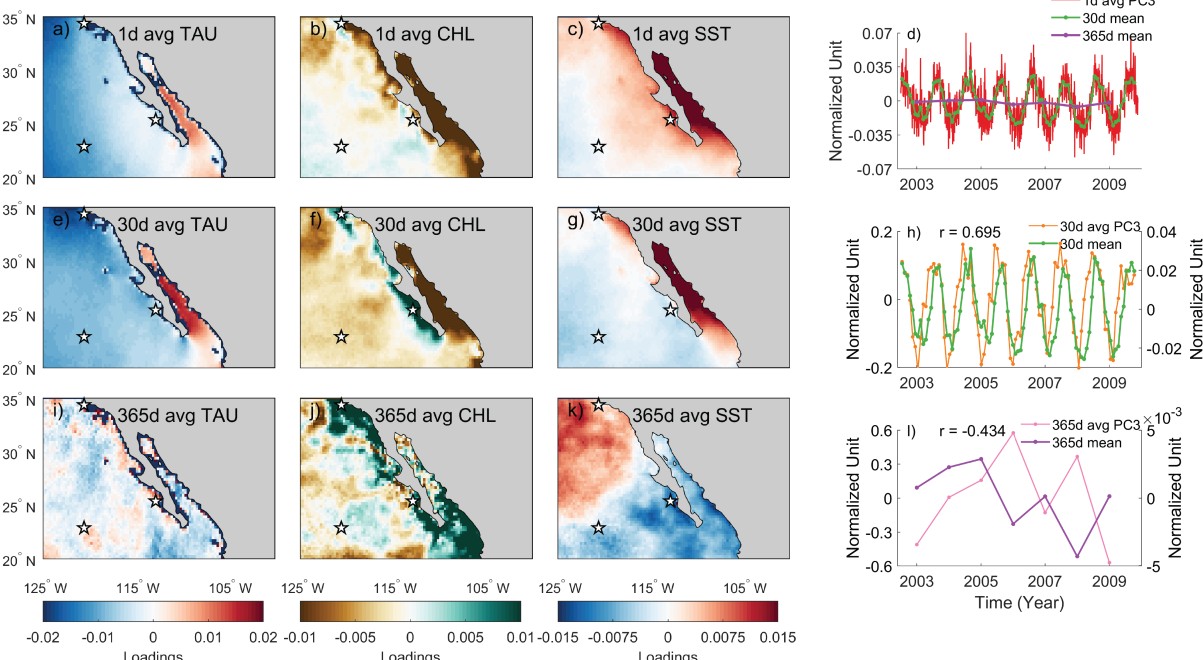

**Figure 7.** EEOF3 spatial and temporal patterns of TAU, CHL, and SST using a–d) daily; e–h) 30 day averaged; i–l) 365 day averaged data. Stars in spatial pattern plots indicate locations where the differences were taken to compute crossshore and meridional gradients. 30 day mean (green) and 365 day mean (purple) timeseries were derived from averaging 1d avg PC3. Correlation coefficient indicates how well does time mean of 1d avg PC track PC of time averaged data.

### 4.3 Are there benefits to analyzing records from multiple sites?

Since an upwelling pattern is only observed in analysis using daily and 30 day averaged data, we focus on assessing the potential benefits to analyzing records from multiple sites on 30 day ($\sim$ monthly) data. We acknowledge that most sedimentary records integrate over annual cycle. However, since we cannot recover upwelling pattern in the first three modes when using 5 365 day averaged data, we here consider an idealized situation instead, where proxy records integrate climate information on $\sim$ monthly timescale.

With only a single proxy type measurement from one site, one can only assume it reflects the dominant large-scale circulation pattern of that area (represented by EEOF1/PC1 in this case). However, comparisons between PC1 and reconstructed PC1s based on a variable from one site shows that the ability to recover the dominant pattern depends on the location and variable 10 (Fig. 8). This varying relationship suggests small-scale processes can drive variability at a proxy site, which can lead to behavior that is different from large-scale circulation. Therefore, caution is needed when trying to extrapolate variability in a single proxy record from one paleoclimate site to infer large-scale circulation changes. Nevertheless, in the absence of additional

sites available to recover sediment cores, we find that measuring multiple variables often lead to better constraint of large scale climate variability (Fig. 9a).

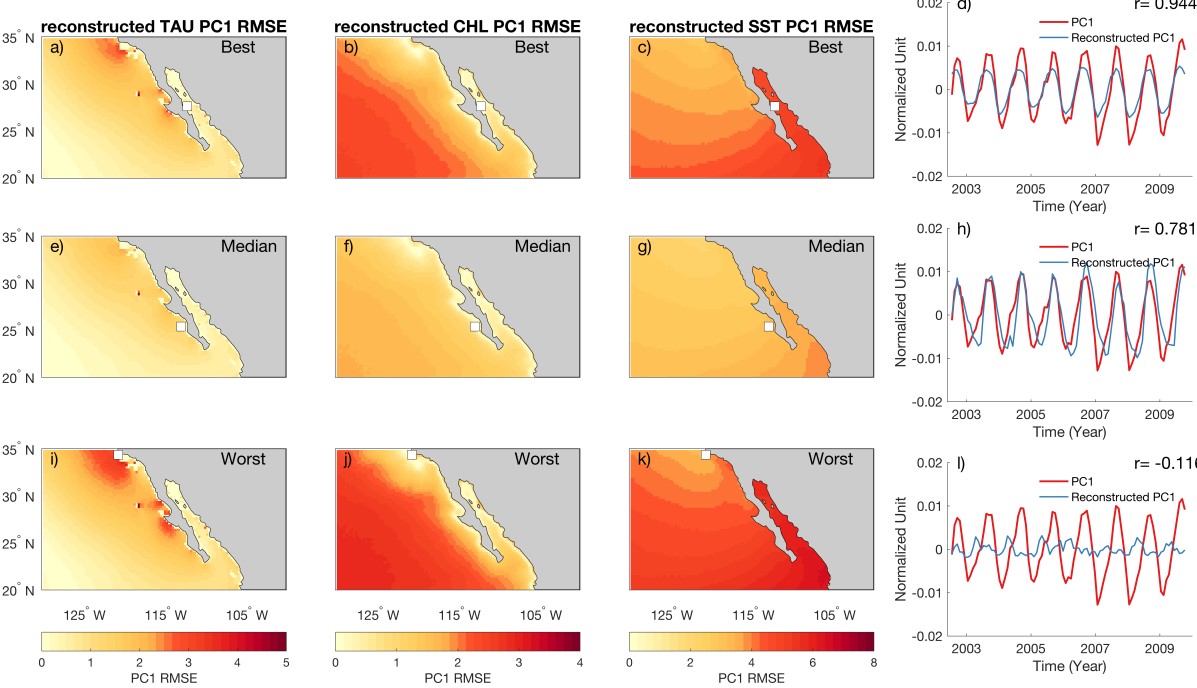

**Figure 8.** (a–d) Best, (e–h) Median, and (i–l) Worst PC1 reconstruction spatial RMSE and timeseries using only 1 variable from 1 site. White marker indicates the site used in that reconstruction, with circle indicating SST and square indicating CHL. The mean of both timeseries were removed for visualization purpose.

Multiple drilling expeditions in SCCS have recovered cores from different locations, which allows us to determine whether there are benefits to analyzing records from multiple sites. With multiple sites available, we can potentially reconstruct different patterns of large scale variability (Figs. 5–7). In the case of 30 day averaged data, a multiple site-based reconstruction allows us to reconstruct spatiotemporal patterns that are associated with Ekman-driven upwelling (Fig. 9). There is also a tendency of increasing reconstruction skill when more sites and proxies are used. Therefore, there is a potential to recover multiple covarying patterns that are driven by different dynamics.

Adding reconstruction sites and variables analyzed can also potentially improve the ability to reconstruct spatiotemporal variability in the spatial domain analyzed. This has been shown in other pseudoproxy experiments that concern hemispheric reconstruction (e.g. Wang et al., 2014). Although our reconstruction technique is rather simple compared to commonly used climate field reconstruction techniques in pseudoproxy experiments and other reconstructions (e.g. Wang et al., 2014), we show that similar results emerge, where increasing number of sites and/or variables can help better reconstruct full field data that

contains multiple variables (Fig. 10; Eq. 3). Therefore, these results together argue for the notion of using multiple sites and proxies for paleoclimate reconstruction.

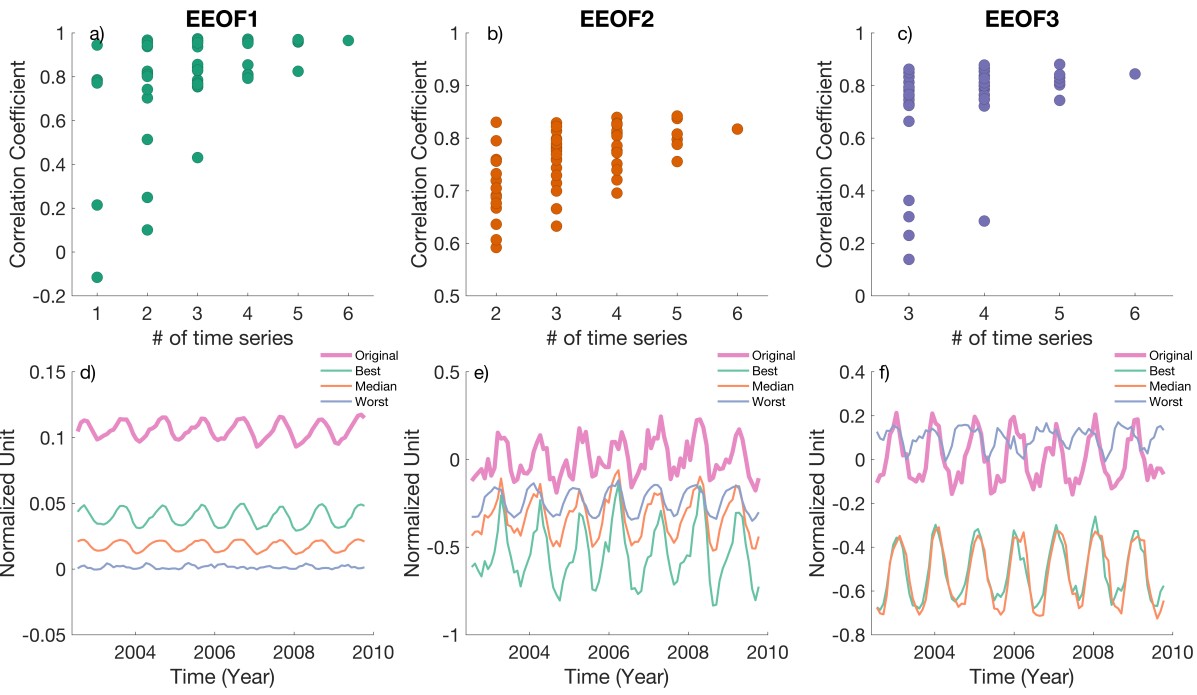

**Figure 9.** Correlation coefficient between reconstructed and actual a) PC1, b) PC2, c) PC3 temporal pattern using 30 day averaged data with varying numbers of time series from the target sites. Also shown are best, median, worst and original temporal pattern reconstructions (ranked by correlation with original PC) of d) PC1, e) PC2, and f) PC3.

## 4.4 Implications

While this study only focuses on the case of Ekman upwelling in SCCS, it has general implications for paleoclimate studies. First, our analysis provides empirical evidence that it is important to consider the spatial representativeness of a proxy record. This calls for careful interpretation in each proxy record developed in order to avoid over simplification and over interpretation of the climate system. Second, we demonstrate that depending on time-averaging and timescale of interest, mechanisms such as Ekman upwelling might/might not be an important process that drives variability in proxy records. Therefore, it is also important to understand whether the proxies applied and the record are able to resolve timescales where the mechanism of interest dominates (e.g. El Nino Southern Oscillation on interannual timescale). Third, we show that analyzing different proxy records from multiple sites can help us reconstruct multiple covarying patterns and improve climate field reconstruction. Last, we propose and demonstrate a multivariate method that allows us to test the assumptions regarding spatial and temporal sampling. We expect that this method can be easily applied to other regions also to provide a first order constraint on how the proxy records can be interpreted.

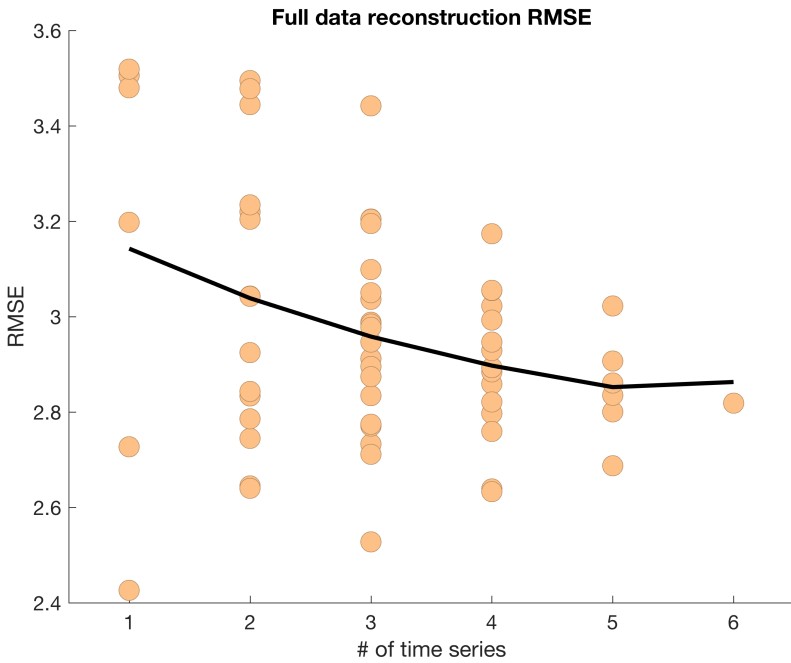

**Figure 10.** Full data reconstruction RMSE using different numbers of timeseries as input for reconstruction

## 4.5 Limitations

There are multiple limitations that have to be taken into account when applying results from this analysis to a paleoclimate context. Firstly, our analysis is only based on 7 years of instrumental data. It is possible that the patterns established in this study are only applicable to the years analyzed due to potential nonstationary covarying relationships between the variables

analyzed. Furthermore, the short length of the instrumental records does not allow us to assess the impacts of basin-scale low-frequency climate variability.

Secondly, our analysis assumes that signals from proxy records can capture surface ocean conditions perfectly and are free from other noise. This assumption is certainly violated, with multiple studies pointing to different sources of uncertainties in sedimentary records (e.g. Dolman and Laepple, 2018). Nevertheless, our analysis provides an idealized scenario to understand

assumptions associated with spatial and temporal sampling, and marks an important step to better interpreting paleoclimate records.

Thirdly, the utilization of chlorophyll satellite product assumes that chlorophyll is related to primary productivity, which in turn is related to export productivity, a variable that is believed to be captured by proxies. While the first assumption that chlorophyll and primary productivity are related is probably accurate on first order (Henson et al., 2010), the relationship

between primary productivity and export productivity is less trivial. Previous studies have identified a general relationship between export productivity, marine productivity and sea surface temperature (Dunne et al., 2005; Laws et al., 2011). Sediment trap studies done in the Santa Barbara and Guaymas basins generally show similar pattern (Thunell et al., 1994; Thunell, 1998),

with export production correlated positively with primary productivity (organic carbon and opal in Santa Barbara Basin; opal in Guaymas Basin). However, discontinuous sediment trap study done in San Lazaro Basin suggested productivity driven by remineralization during El Nino, which resulted a low export productivity despite high productivity (Silverberg et al., 2004). This highlights the potential complexity in plankton communities along a continental margin, which can experience both eutrophic and oligotrophic conditions. In fact, Dunne et al. (2005) examined the proposed parameterization by synthesizing different sediment trap sites and showed that the positive relationship between primary productivity and export productivity works in a global sense but not small scales. Furthermore, many studies have highlighted other factors to consider when considering export production, for instance particle size, ballasting effects, remineralization, eddy subduction, mixed layer pumping (Lam and Marchal, 2015; Boyd et al., 2019, and references therin). Hence, more dedicated experiments are needed in order to establish a quantitative relationship between chlorophyll data used here and paleo-productivity records.

Fourthly, we assume that each statistical mode retrieved in this study is tied to a dynamical mechanism. However, previous studies have cautioned against such interpretations (e.g. Hannachi et al., 2007). Nevertheless, our study does not aim to diagnose Ekman upwelling processes but simply aims to determine whether it is possible to recover Ekman upwelling related patterns in proxy records. Hence, we argue the distinction between a dynamical mode and statistical mode does not undermine our results.

Lastly, our multiple record analysis assumes proxy records contain perfect age models. In most cases, this assumption is also invalid. It is inevitable that each sedimentary record contains absolute age uncertainties. Therefore, using marine sedimentary records for a multi-site proxy reconstruction with a high-temporal resolution is more challenging and might yield a different conclusion than ours.

## 5   Conclusions

This study aimed to evaluate assumptions commonly made in paleoclimate studies – (1) certain mechanism operates in the past on all timescales of interest, and (2) large-scale phenomena can explain the most variance in a small location (i.e. a paleoclimate site). We tested these assumptions by focusing on the Southern California Current System and used observational records to understand whether it is possible to reconstruct Ekman Upwelling using multiple sedimentary records. We introduced an Extended Empirical Orthogonal Function framework and applied it to satellite records to make inferences about paleoclimate records. Our results indicate the dominant TAU, CHL, SST covarying pattern does not resemble Ekman upwelling. In addition, the relationship between these variables appears to depend on timescales and spatial scales. A positive result is that our analysis suggests that a few sediment sites can monitor large scale fields associated with the Southern California Current. Lastly, we highlight the potential benefits of using multiple proxy records to understand different large scale covarying patterns. Our study suggests that instrumental records are helpful for testing assumptions in paleoclimatology, and the associated spatial and temporal-scale extrapolations made based on paleoclimate reconstructions. Testing these assumptions might help us better interpret proxy records and understand past climate changes.

*Code availability.* Data and MATLAB scripts used to produce figures of this manuscript are available at: https://doi.org/10.26300/41y9-ts23

*Author contributions.* AHC, BFK, TDH developed the idea. AHC and BFK designed the analysis. AHC carried out the analysis and wrote the manuscript with inputs from BFK and TDH.

*Competing interests.* The authors declare no conflict of interest

5   *Acknowledgements.* The Geostationary Environmental Satellites SST, QuikSCAT surface wind data were obtained from the NASA EOSDIS Physical Oceanography Distributed Active Archive Center (PO.DAAC) at the Jet Propulsion Laboratory, Pasadena, CA. MODIS chlorophyll-a data were obtained from NASA Goddard Space Flight Center, Ocean Ecology Laboratory, Ocean Biology Processing Group. We thank R. Vachula, N. Richter, S. Garelick for helpful comments, suggestions and discussions. AHC was supported by the Brown University Presidential Fellowship. BFK was supported by ONR N00014-17-1-2393.

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
