# Peer review of "Can we use sea surface temperature and productivity proxy records to reconstruct Ekman Upwelling?"

_Climate of the Past, 2019_

## Referee Comment (RC1) · Anonymous Referee #1 · 13 Aug 2019

The paper by Cheung et al. uses satellite data to test whether Ekman pumping is likely to be detectible in marine sediment records from the Southern Californian margin through proxy records, especially those relating to SST and productivity. They conclude that important processes, such as Ekman pumping, do not occur across all timescales, and that an integrated proxy record is unlikely to accurately reflect the spatial variability associated with Ekman pumping. They further show that inclusion of multiple sites may increase the reliability of proxy records. Overall, this is an interesting study and the conclusions seem sound and well supported. I do have a few comments, which I hope can serve to improve the manuscript.

[Figure]

First, I am surprised the authors have not included much of a temporal element in applying their modern observations to sediment core reconstructions. The authors are quite convincing in showing that annually averaged (or seasonally weighted) models do not represent the true spatial extent of Ekman upwelling. However, what they don't seem to test is whether time averaged variability in these phenomena would be captured in the sediment. After all, this is what the vast majority of marine sediment studies record – temporal variability (even relative) at a site, rather than comparison between sites in a world with perfect age models. I think including a test of how integrated variables compare across different intervals would be relevant to addressing this point.

Second, even after extremely close reading, I am struggling to understand how exactly the pseudo-proxy time-series presented in Figures 8-10 were generated. I'm under the impression that each of these is an integration of satellite data at a specific point. Is this correct? If so, this could be made more explicit, and an explanation of how and why particular sites were chosen would be helpful.

Minor comments: Page 2, Line 3 – Missing the end of this sentence.

Figure 8 – Something is going on here with the labeling of "best", "median", and "worst." I don't think this is correct.

---

## Referee Comment (RC2) · Anonymous Referee #2 · 1 Sep 2019

In this study, the authors conducted statistical analysis on the co-variability among SST, wind stress, and chlorophyll (CHL) concentration using satellite-derived high spatial and temporal resolution data. The study is focused on an Eastern Boundary Upwelling Systems (EBUS), the southern California region. The results suggest that the dominant mode of co-variability among SST, wind stress and CHL does not reflect the Ekman upwelling process. The second and third modes of co-variability are found to reflect upwelling patterns but exhibit complicated region and timescale dependence. The authors findings imply that paleoclimate records over the EBUS may reflect complicated physical processes other than the Ekman upwelling.

[Figure]

The scope of the study is of course important as there is large uncertainty in the interpretation of paleoclimate reconstructions and the related implications are huge. However, I think the limitation from short length of data could be better explored. Physical process-based interpretations of statistical results should be better presented. Please see my detailed comments bellow.

Major comments:

1. Page 9 Line 19: "Instead of strong cross-shore gradients, TAU and CHL display a weak cross-shore gradient, and SST exhibits a meridional gradient pattern (Fig. 5)." Can you be more quantitative? How is the gradient defined? How big are the gradients to be considered strong or weak?

2. What is the physical meaning of EEOF1 in Figure 5? Are these statistical results physically meaningful? The authors briefly described the spatial pattern but did not show any results on the temporal variability (PCs). What do the PCs tell us about the co-variability among these variables?

3. Limitation from short instrumental data could be better explored. Length of data analyzed in the manuscript is only seven years, which could limit interpretations of the results, especially considering the interannual and decadal variability in the system. The co-variability between SST and TAU could be explored in high-resolution ocean reanalysis, e.g., the Simple Ocean Data Assimilation (SODA; http://www.soda.umd.edu/). SODA provides SST and wind stress data of $\frac{1}{4}°$-horizontal and 5-day temporal resolution with a length of 30+ years. The spatial and temporal resolution of SODA is comparable to the data sets in the authors' manuscript, but the data length is much longer. With longer data coverage, we can, at least, examine (1) whether the dominant co-varying pattern between TAU and SST reflects Ekman upwelling, (2) whether results from satellite data are consistent with reanalysis, (3) how the results depend on low-frequent climate variability, and (4) how CHL variation may further complicate the co-variability among these variables.

[Figure]

Minor comments: 1. What is the variance explained by each EEOF mode?

---

## Referee Comment (RC3) · Yige Zhang (Referee) · 3 Sep 2019

Cheung et al. analyzed modern satellite observations to address the question of whether records preserved in marine sediments can be used to reconstruct Ekman Upwelling in Earth's history. This paper is well written. Deep-water upwelling is important in regulating global climate and biogeochemical cycles, and understanding the modern, instrumental records is paramount for paleo-applications. The take home message is that multi-site and multi-variable reconstructions are the preferred way of evaluation ancient upwelling. I'd like to see more studies like this published on Climate of the Past, and recommend publication of this manuscript after addressing a few

comments.

I understand that the available satellite-based observations include SSTs, chlorophyll-a
and alongshore wind stress. And the authors realized that these factors do not directly
translate into proxy-derived information ("Although CHL does not equate precisely to
primary productivity, and also differs from productivity inferred from proxy records"), I'd
appreciate more elaborations on how to build connections between these two types of
variables. Anyhow, this is a ms for Climate of the Past, and the audience would want
to know.

For example, SST would be less of a problem. But common proxies for productivity
(e.g., Ba, opal accumulation etc) are actually looking at export productivity. How are
they expected to be different from CHL data and are they better in tracking upwelling?
Also, I know that one paper cannot address everything, but recent studies have sug-
gested that the carbon cycle might be more sensitive than SSTs to equatorial upwelling
(Keller et al., 2015, GRL). Zhang et al., (2017, EPSL) used air-sea disequilibria of $CO_2$
and export production to infer deep-water upwelling in the eastern equatorial upwelling,
which reached very different conclusions from the SST results. Can this modern study
weigh in to help people disentangle what is "upwelling" and what is not from the sedi-
ment data?

There are a few other issues. For examples, I'm also confused like the other Referee
about how Fig. 8-10, the pseudo-proxy time-series were generated.

---

## Author Comment (AC1) · 18 Sep 2019

**Reviewer #1**

The paper by Cheung et al. uses satellite data to test whether Ekman pumping is likely to be detectible in marine sediment records from the Southern Californian margin through proxy records, especially those relating to SST and productivity. They conclude that important processes, such as Ekman pumping, do not occur across all timescales, and that an integrated proxy record is unlikely to accurately reflect the spatial variability associated with Ekman pumping. They further show that inclusion of multiple sites may increase the reliability of proxy records. Overall, this is an interesting study and the conclusions seem sound and well supported. I do have a few comments, which I hope can serve to improve the manuscript.

We thank the reviewer for pointing out sections that require clarification and providing constructive comments. Point to point reply to comments are below.

First, I am surprised the authors have not included much of a temporal element in applying their modern observations to sediment core reconstructions. The authors are quite convincing in showing that annually averaged (or seasonally weighted) models do not represent the true spatial extent of Ekman upwelling. However, what they don't seem to test is whether time averaged variability in these phenomena would be captured in the sediment. After all, this is what the vast majority of marine sediment studies record – temporal variability (even relative) at a site, rather than comparison between sites in a world with perfect age models. I think including a test of how integrated variables compare across different intervals would be relevant to addressing this point.

Although we did not address directly the issue of proxy averaging because averaging windows of proxy records are typically longer than the observational record used in this study, we attempted to show the effects of timescale averaging on large scale circulation (defined by the EOF patterns of the study region) by comparing spatial patterns using daily averaged, 30 day averaged, and 365 day averaged data (Figs. 5-7). A separate concern for sediments, if the proxy records are not a pure average, but an average weighted toward particular seasons (e.g., based on productivity variability or sedimentation rates by season) is occurring is a highly complex and proxy-dependent discussion. Taking that into account fully is far beyond the scope of this paper. However, we now include a more detailed discussion on the implications of proxy interpretations based on the results we present, specifically pointing out how different patterns seen when using different time averaged data imply the need to interpret proxies differently depending on the timescale proxies average (e.g. seasonal vs annual) and whether those averages include non-uniform weighting.

Second, even after extremely close reading, I am struggling to understand how exactly the pseudo-proxy time-series presented in Figures 8-10 were generated. I'm under the impression that each of these is an integration of satellite data at a specific point. Is this correct? If so, this could be made more explicit, and an explanation of how and why particular sites were chosen would be helpful.

We agree that this was not very clear. We have remade the figures and we are also adding a more detailed description on how Figures 8-10 were generated, providing the explicit algorithms and equations.

Minor comments:

Page 2, Line 3 – Missing the end of this sentence.

Thanks for pointing it out. The sentence is now completed in the updated manuscript.

Figure 8 – Something is going on here with the labeling of "best", "median", and "worst." I don't think this is correct.

We went back and double checked the script that generate this figure. We realized there was a mistake in computing the scaling factor, which has been corrected. Furthermore, we double-checked the selection criterion for "best", "median", and "worst" and made sure that the correct instances are being selected and presented here.

---

## Author Comment (AC2) · 18 Sep 2019

In this study, the authors conducted statistical analysis on the co-variability among SST, wind stress, and chlorophyll (CHL) concentration using satellite-derived high spatial and temporal resolution data. The study is focused on an Eastern Boundary Upwelling Systems (EBUS), the southern California region. The results suggest that the dominant mode of co-variability among SST, wind stress and CHL does not reflect the Ekman upwelling process. The second and third modes of co-variability are found to reflect upwelling patterns but exhibit complicated region and timescale dependence. The authors findings imply that paleoclimate records over the EBUS may reflect complicated physical processes other than the Ekman upwelling. The scope of the study is of course important as there is large uncertainty in the interpretation of paleoclimate reconstructions and the related implications are huge. However, I think the limitation from short length of data could be better explored. Physical process-based interpretations of statistical results should be better presented. Please see my detailed comments bellow.

We thank the reviewer for pointing out sections that require clarification and providing constructive comments. Point to point reply to comments are below.

Major comments:

1. Page 9 Line 19: "Instead of strong cross-shore gradients, TAU and CHL display a weak cross-shore gradient, and SST exhibits a meridional gradient pattern (Fig. 5)." Can you be more quantitative? How is the gradient defined? How big are the gradients to be considered strong or weak?

Yes. We now take 3 points that are approximately meridional and cross-shore (will be marked as stars in figures 5-7 in the revised manuscript) and compute the difference divided by its arc length to define the gradient. The strength of the gradients is defined in a relative sense by comparing cross shore, meridional gradients and gradients using different time average. Based on this definition of gradients, we have refined our discussion quantifying the changing patterns.

2. What is the physical meaning of EEOF1 in Figure 5? Are these statistical results physically meaningful? The authors briefly described the spatial pattern but did not show any results on the temporal variability (PCs). What do the PCs tell us about the co-variability among these variables?

We believe the EEOF1 pattern shown represents annual cycle in conditions of SST, CHL, and TAU. They are statistically meaningful in the sense that they optimize the compaction of data according to the standard EEOF approach, but physical meaning is not a requirement of the EEOF method so this interpretation is only suggestive of the physical process. We do show the 30 day averaged PCs1-3 in Figure 8-9. We now include PCs in Figures 5-7 as well. The PCs describe how these covarying patterns (each EEOF) change over time. We expand our discussion of the meanings of EEOF and PC in the methods section to clarify their physical meaning.

3. Limitation from short instrumental data could be better explored. Length of data analyzed in the manuscript is only seven years, which could limit interpretations of the results, especially considering the interannual and decadal variability in the system. The co-variability between SST and TAU could be explored in high-resolution ocean reanalysis, e.g., the Simple Ocean Data Assimilation (SODA; http://www.soda.umd.edu/). SODA provides SST and wind stress data of 1 4 ∘ -horizontal and 5-day temporal resolution with a length of 30+ years. The spatial and temporal resolution of SODA is

comparable to the data sets in the authors' manuscript, but the data length is much longer. With longer data coverage, we can, at least, examine (1) whether the dominant co-varying pattern between TAU and SST reflects Ekman upwelling, (2) whether results from satellite data are consistent with reanalysis, (3) how the results depend on low-frequent climate variability, and (4) how CHL variation may further complicate the co-variability among these variables.

We agree that SODA data covers a longer period and has resolution comparable to the interpolated data used in this study and can potentially provide additional information about longer term climate variability. However, many other issues can arise if we use SODA or any other model for analysis. Firstly, although SODA is modeled at 0.25 x 0.25 resolution, the reanalysis assimilates observations at 1 x 1 resolution (Carton et al. 2018). Hence, SODA relies on the ocean model to simulate <1º resolution properties, and modeling of any features smaller than 0.25 degrees are parameterized. Secondly, although the GFDL CM2.5 model is eddy permitting, it is only mesoscale eddy permitting. Yet, previous studies have suggested abundant submesoscale fronts, eddies, and other features inhabit and dominate short term variability of upwelling systems, which can affect spatial structure of sea surface temperature and marine productivity, and a minimum resolution to simulate these features is in the 500m to 1km range (Capet et al. 2008), 25 times higher than in SODA. Using a mesoscale-permitting model such as SODA means that parameterizations are expected to represent these phenomena. Satellite data, while limited in sampling resolution, of course relies on the true biophysical system behavior not a simulated version of it. Thirdly, the quality of reanalysis depends on initialization, surface forcing, open boundary conditions, model physics, and measurement biases. All these factors come with their own associated uncertainties. At the moment, we use only satellites and EEOFs, and the discussion of those uncertainties is already a large fraction of the paper. Hence, the number of additional uncertainties that need to be discussed and taken account of when using reanalysis products is considerably larger than compared to satellite data for the specific process of upwelling. Therefore, we decided not to pursue the path of using reanalysis product. We will include a brief discussion on the rationale of using exclusively satellite data in the revised manuscript.

Minor comments: 1. What is the variance explained by each EEOF mode?

By analyzing the singular values, EEOF1 explains ~84% of the total variance, EEOF2 and 3 each explains ~5% of the total variance. This is not mentioned explicitly in the text.

References:

Capet, X., McWilliams, J. C., Molemaker, M. J., & Shchepetkin, A. F. (2008). Mesoscale to submesoscale transition in the California Current System. Part I: Flow structure, eddy flux, and observational tests. *Journal of physical oceanography*, *38*(1), 29-43.

Carton, J. A., Chepurin, G. A., & Chen, L. (2018). SODA3: A new ocean climate reanalysis. *Journal of Climate*, *31*(17), 6967-6983.

---

## Author Comment (AC3) · 18 Sep 2019

Cheung et al. analyzed modern satellite observations to address the question of whether records preserved in marine sediments can be used to reconstruct Ekman Upwelling in Earth's history. This paper is well written. Deep-water upwelling is important in regulating global climate and biogeochemical cycles, and understanding the modern, instrumental records is paramount for paleo-applications. The take home message is that multi-site and multi-variable reconstructions are the preferred way of evaluation ancient upwelling. I'd like to see more studies like this published on Climate of the Past, and recommend publication of this manuscript after addressing a few comments.

We thank the reviewer for pointing out sections that require clarification and providing constructive comments. Point to point reply to comments are below.

I understand that the available satellite-based observations include SSTs, chlorophyll-a and alongshore wind stress. And the authors realized that these factors do not directly translate into proxy-derived information ("Although CHL does not equate precisely to primary productivity, and also differs from productivity inferred from proxy records"), I'd appreciate more elaborations on how to build connections between these two types of variables. Anyhow, this is a ms for Climate of the Past, and the audience would want to know. For example, SST would be less of a problem. But common proxies for productivity (e.g., Ba, opal accumulation etc) are actually looking at export productivity. How are they expected to be different from CHL data and are they better in tracking upwelling? Also, I know that one paper cannot address everything, but recent studies have suggested that the carbon cycle might be more sensitive than SSTs to equatorial upwelling (Keller et al., 2015, GRL). Zhang et al., (2017, EPSL) used air-sea disequilibria of $CO_2$ and export production to infer deep-water upwelling in the eastern equatorial upwelling, which reached very different conclusions from the SST results. Can this modern study weigh in to help people disentangle what is "upwelling" and what is not from the sediment data?

We agree that it is important to try to build a relationship between export productivity (or marine productivity proxy records) and chlorophyll satellite data as this could help putting results from our study into the context of paleoclimate reconstruction. However, we believe it is currently not possible to identify a reasonable quantitative relationship between primary productivity and export productivity and outside the scope of our study. Our argument is as follow.

Previous studies have identified a general relationship between export productivity, marine productivity and sea surface temperature (Dunne et al. 2005; Laws et al. 2011). Sediment trap studies done in the two basins mentioned in this study generally show similar pattern (Thunell et al. 1994; Thunell 1998), with export production correlated positively with primary productivity (organic carbon and opal in Santa Barbara Basin; opal in Guaymas Basin). However, discontinuous sediment trap study done in San Lazaro Basin also suggested productivity driven by remineralization during El Nino, which resulted a low export productivity despite high productivity (Silverberg et al. 2004). This highlights the potential complexity in continental margin, where they can experience both eutrophic and oligotrophic conditions. In fact, Dunne et al. (2005) examined the proposed parameterization by synthesizing different sediment trap sites and showed that the positive relationship between primary productivity and export productivity works in a global sense but not small scales. Furthermore, many studies have highlighted other factors to consider when considering export production, for instance particle size, ballasting

effects, remineralization, eddy subduction, mixed layer pumping (see Lam and Marchal 2015, Boyd et al. 2019 and references therein). While it might be possible to infer particle size in a paleoclimate context based on measurements of different proxies (nannofossil assemblages, comparison between diatom productivity proxy and coccolithophore productivity proxy), other parameters cannot be constrained in proxy records. Therefore, we believe it is currently not possible to argue for a quantitative relationship between primary productivity and export productivity. Nevertheless, we now include a brief discussion on the relationship between primary productivity and export productivity at our sites.

There are a few other issues. For examples, I'm also confused like the other Referee about how Fig. 8-10, the pseudo-proxy time-series were generated

We agree that this was not very clear. We have remade the figures and added a detailed description including the algorithm and equations on how Figures 8-10 were generated.

References:

Boyd, P. W., Claustre, H., Levy, M., Siegel, D. A., & Weber, T. (2019). Multi-faceted particle pumps drive carbon sequestration in the ocean. *Nature*, *568*(7752), 327.

Dunne, J. P., Armstrong, R. A., Gnanadesikan, A., & Sarmiento, J. L. (2005). Empirical and mechanistic models for the particle export ratio. *Global Biogeochemical Cycles*, *19*(4).

Lam, P. J., & Marchal, O. (2015). Insights into particle cycling from thorium and particle data. *Annual review of marine science*, *7*, 159-184.

Laws, E. A., D'Sa, E., & Naik, P. (2011). Simple equations to estimate ratios of new or export production to total production from satellite-derived estimates of sea surface temperature and primary production. *Limnology and Oceanography: Methods*, *9*(12), 593-601.

Silverberg, N., Martínez, A., Aguíñiga, S., Carriquiry, J. D., Romero, N., Shumilin, E., & Cota, S. (2004). Contrasts in sedimentation flux below the southern California Current in late 1996 and during the El Niño event of 1997–1998. *Estuarine, Coastal and Shelf Science*, *59*(4), 575-587.

Thunell, R. C., Pride, C. J., Tappa, E., & Muller-Karger, F. E. (1994). Biogenic silica fluxes and accumulation rates in the Gulf of California. *Geology*, *22*(4), 303-306.

Thunell, R. C. (1998). Particle fluxes in a coastal upwelling zone: sediment trap results from Santa Barbara Basin, California. *Deep Sea Research Part II: Topical Studies in Oceanography*, *45*(8-9), 1863-1884.